# Definition of IgG Subclass-Specific Glycopatterns in Idiopathic Membranous Nephropathy: Aberrant IgG Glycoforms in Blood

**DOI:** 10.3390/ijms23094664

**Published:** 2022-04-23

**Authors:** Clizia Chinello, Noortje de Haan, Giulia Capitoli, Barbara Trezzi, Antonella Radice, Lisa Pagani, Lucrezia Criscuolo, Stefano Signorini, Stefania Galimberti, Renato Alberto Sinico, Manfred Wuhrer, Fulvio Magni

**Affiliations:** 1Clinical Proteomics and Metabolomics Unit, School of Medicine and Surgery, University of Milano-Bicocca, 20854 Vedano al Lambro, Italy; lisa.pagani@unimib.it (L.P.); l.criscuolo@campus.unimib.it (L.C.); fulvio.magni@unimib.it (F.M.); 2Center for Proteomics and Metabolomics, Leiden University Medical Center, 2333 ZA Leiden, The Netherlands; ndehaan@sund.ku.dk (N.d.H.); m.wuhrer@lumc.nl (M.W.); 3Copenhagen Center for Glycomics, University of Copenhagen, 1165 Copenhagen, Denmark; 4Bicocca Bioinformatics Biostatistics and Bioimaging B4 Center, Department of Medicine and Surgery, University of Milano-Bicocca, 20900 Monza, Italy; giulia.capitoli@unimib.it (G.C.); stefania.galimberti@unimib.it (S.G.); 5Department of Medicine and Surgery, University of Milano Bicocca and Nephrology, 20900 Monza, Italy; barbara.trezzi@unimib.it (B.T.); renato.sinico@unimib.it (R.A.S.); 6Dialysis Unit, ASST-Monza, Ospedale San Gerardo, 20900 Monza, Italy; 7Microbiology Institute, ASST (Azienda Socio Sanitaria Territoriale) Santi Paolo e Carlo, 20142 Milan, Italy; antonella.radice@asst-santipaolocarlo.it; 8Department of Laboratory Medicine, Hospital of Desio, 20832 Desio, Italy; s.signorini@asst-monza.it

**Keywords:** IMN, IgG, LC-MS, N-glycans, glycosylation, galactosylation, glycoproteomics

## Abstract

The podocyte injury, and consequent proteinuria, that characterize the pathology of idiopathic membranous nephropathy (IMN) is mediated by an autoimmune reaction against podocyte antigens. In particular, the activation of pathways leading to abundant renal deposits of complement is likely to involve the binding of mannose-binding lectin (MBL) to aberrant glycans on immunoglobulins. To obtain a landscape of circulatory IgG Fc glycosylation characterizing this disease, we conducted a systematic N-glycan profiling study of IgG1, 2, and 4 by mass spectrometry. The cohort included 57 IMN patients, a pathological control group with nephrotic syndrome (PN) (*n* = 20), and 88 healthy control subjects. The effect of sex and age was assessed in all groups and controlled by rigorous matching. Several IgG Fc glycan traits were found to be associated with IMN. Interestingly, among them, only IgG4-related results were specific for IMN and not for PN. Hypo-galactosylation of IgG4, already shown for IMN, was observed to occur in the absence of core fucose, in line with a probable increase of pro-inflammatory IgG. In addition, elevated levels of fucosylated IgG4, along with low levels of hybrid-type glycans, were detected. Some of these IgG4 alterations are likely to be more pronounced in high PLA2R (phospholipase A2 receptor) patients. IgG Fc glycosylation patterns associated with IMN warrant further studies of their role in disease mechanisms and may eventually enrich the diagnostic spectrum regarding patient stratification.

## 1. Introduction

Membranous nephropathy (MN) is a glomerular disease with a relatively low incidence, being responsible for approximately 20% of all cases nephrotic syndrome worldwide, but with a high propensity to progress to end-stage renal disease [1]. It is histopathologically recognizable by an apparent thickening of the capillary walls that is due to the deposition or formation of immune complexes in the glomerular basement membrane. The antigens initiating this complex formation may be part of the basement membrane or deposited from elsewhere following their circulation [2].

MN can present a primary (idiopathic; IMN) or secondary etiology. In developed countries, approximately 75% of all MN are idiopathic, whilst the remaining 20–25% are secondary to different conditions, such as infections (hepatitis B virus, hepatitis C virus, human immunodeficiency virus, malaria, etc.), malignancies (lung, breast, stomach, and ovarian cancer; lymphoproliferative disorder, etc.), systemic autoimmune disease (lupus, rheumatoid arthritis, etc.), and drug toxicity [3]. Diagnosis of both primary and secondary MN relies on renal biopsy, in particular on the finding of deposits of IgG in a granular pattern along the glomerular capillary loop by immunofluorescence. Some histological features may help to discriminate between the two forms of MN. In IMN, no endocapillary proliferation is usually seen and the presence of a pronounced mesangial proliferation implies a secondary form of MN [4]. For unknown reasons, IgG4 is the dominant autoantibody isotype in IMN. A positive staining for non-IgG4 subclasses (IgG1, IgG3), IgA, IgM, or significant staining in the glomerular mesangium is suggestive of secondary MN, often lupus nephritis class V [5]. The distinction between primary and secondary forms also has crucial therapeutic implications given that immunosuppressive drugs are administered in cases of IMN and that such treatment is harmful in membranous nephropathy, secondary to hepatitis C or hepatitis B virus infections, that have to be treated by antiviral drugs [1,5].

In 2009, Beck et al. demonstrated the presence of circulating anti-PLA2R (phospholipase A2 receptor) antibodies in 26 of 37 IMN patients, performing Western blotting with extracts from normal human glomeruli as a source of antigens [6]. The anti-PLA2R antibodies were highly specific for IMN as they were not detected neither in the sera of controls nor patients affected by other primary glomerulonephritis or by secondary forms of MN [7]. Very recently, a number of new target antigens have been identified in membranous nephropathy and these include, in addition to PLA2R, exostosin 1/2, neural epidermal growth-like 1 protein, semaphorin 3B, protocadherin 7, neural cell adhesion molecule 1, protocadherin FAT1, and thrombospondin type 1 domain containing 7A (THSD7A). Some of these antigens are present in the setting of IMN, some in secondary MN and some in both. The utility of these newly discovered antigen-specific antibodies has been suggested for the classification MN, but they have not yet been introduced in clinical practice as the data is still preliminary [8]. Therefore, as also published in the recent KDIGO (Kidney Disease Improving Global Outcomes) guidelines, it is recommended that patients with MN should be evaluated for associated conditions, regardless of whether concerning anti-PLA2R antibodies and/or anti-THSD7A [9].

Circulating anti-PLA2R antibodies of patients with IMN are predominantly, but not exclusively, of the IgG4 subclass. PLA2R and IgG4 have been shown to co-localize in the immune deposits in IMN patients at the subepithelial level and only the IgG eluted from the biopsy samples of IMN patients reacted with the PLA2R [10,11]. Circulating autoantibodies to PLA2R were detected in approximately 70% of patients with IMN in all the different studies performed [10]. Even if there is a consensus of using anti-PLA2R antibodies as a diagnostic and prognostic biomarker, many issues addressing initiation of antibody formation and the timeline of their production along with the role of IgG subclasses and antigenic epitopes, remain unsolved [12]. MN can indeed be considered an organ-specific autoimmune disease. What drives autoimmunity is not known. PLA2R-associated IMN most likely develops through the interaction of factors such as genetic susceptibility, loss of tolerance, alterations in antigen expression with a role for environmental factors like air pollution, smoking, and infections. Genome-wide association studies have shown highly significant associations of the 6p21 HLA- DQA1 (Major Histocompatibility Complex, Class II, DQ Alpha 1) and 2q24 PLA2R1 loci with idiopathic membranous nephropathy in Caucasian European patients. These risk alleles of the two genes had an additive effect for development of IMN. Patients carrying all four risk alleles had an odds ratio close to 80, compared with individuals who had only the protective alleles [5]. These observations, together with preliminary studies showing that anti-PLA2R IgG4 can bind to mannose-binding lectin (MBL), a major component of the lectin complement pathway, suggests that both the lectin complement pathway and IgG4 may be involved in the pathogenesis of the disease [13]. Interestingly, the activation of this pathway is likely to involve the binding of MBL to the aberrant glycans present on immunoglobulins [14]. In a cohort of IMN patients, Haddad et al. identified synaptopodin and NEPH1, two essential podocyte proteins, as the targets of complement-induced podocyte injury. An altered glycosylation pattern of IgG4 was identified in IMN patients and this facilitated complementing activation through the lectin pathway [15].

Fc glycans of IgG are important modulators of effector functions [16,17,18,19]. Changes in this system are generally associated with different inflammatory and autoimmune diseases [20], including kidney disorders [21]. For example, it has been shown that the deposition and formation of the immunocomplex IgA–IgG in patients affected by IgA nephropathy could be ascribed to the aberrant glycosylation of IgA1 [22]. Moreover, several studies utilizing animal models have shown that modulation of IgG glycosylation is a promising strategy to interfere with the anti-neutrophil cytoplasm autoantibody (ANCA)-mediated inflammatory processes associated with glomerulonephritis [23], in addition to that Ig glycan analysis can contribute to the prediction of the relapse in a defined cohort of autoimmune vasculitis patients [24].

Mass spectrometric techniques have advanced considerably in recent years and this has facilitated the in-depth characterization of alterations in the glycoportion of the IgG present in human blood in a subclass-specific manner, revealing associations with a plethora of human diseases, in particular those related to inflammation and metabolic health [25]. It has been observed that these associations can involve all subclasses, or may be subclass-specific [26]. For example, it has been reported that the galactosylation and sialylation patterns of IgG1 and IgG4, with respect to IgG2, are more closely associated with metabolic markers, even whilst the direction of these correlations were consistent for all the different IgG subclasses [26]. The IgG N-glycome of blood is also strongly affected by sex and age, and these can act as critical confounding factors in biomarker discovery [27]. A study from Baković et al. carried out in more than 1700 subjects showed that this dependence was more evident in younger individuals (<57 years) than in older individuals (>57 years) and in females than in males [28]. The diagnostic potential of different IgG glycoforms has been demonstrated to be both sex- and age-dependent in various pathologies such as human lung cancer [29] and rheumatoid arthritis [30].

Although a significant increase in serum levels of IgG4 has already been observed in IMN [31], no systematic age- and sex-matched glycoprofiling study has been conducted on IgG forms in blood of patients affected by IMN. Here, we present a mass spectrometric glycoproteomic characterization of IgG subclasses in the serum of anti-PLA2R-pos IMN patients (*n* = 57) and in subjects affected by different nephrotic syndromes (*n* = 20). To improve the specificity of the disease signatures, the effect of sex and age was assessed via a strict double matching for all the comparisons within a selected group of healthy controls (*n* = 88).

## 2. Results

### 2.1. Age and Sex Effect on IgG Fc Glycosylation in the Cohort

Variations in IgG glycosylation can strongly impact effector function and can be related to the pathological status of an individual. However, as reported in the literature, changes are often also associated with other factors such as age and sex, as well as the disease [27]. For instance, the IgG Asn297-linked glycan has been found to be hypo-sialylated and hypo-galactosylated, with elevated levels of glycans terminating with GlcNAc (N-acetylglucosamine), in many inflammatory and autoimmune conditions, as well as with ageing [32,33]. For these reasons, a prior evaluation of the effect of age and sex on IgG Fc glycosylation in our cohort of patients and controls was performed to avoid possible artefacts in the search for molecular signatures of IMN.

The study comprised 57 IMN patients and 20 pathological controls with other types of nephropathy (characteristics are summarized in Appendix A). Since PLA2R is the predominant autoantigen in adults with membranous kidney disease and circulating anti-PLA2R antibodies are found in 70–80% of patients [34], the subjects were subdivided into high PLA2R (*n* = 29) or low PLA2R (*n* = 28) individuals. None of the PN patients showed PLA2R Ab (AntiBodies) titers (<10). The age distribution of the included healthy controls (CTRLs; *n* = 88) was comparable to the disease groups (17 to 87 years for CTRLs and 19 to 81 years for IMN+PN) (Figure 1). There was an unbalanced age distribution between PN and IMN patients, with a higher age being observed in the IMN group. About 35% of the healthy subjects and of the combined patients with a nephropathic disease were females, whilst the PN group showed a higher incidence of females (50%) with respect to the IMN group (26%) (Appendix A). Due to the discrepancies in terms of age and sex between PN and IMN, a direct comparison between these two groups was avoided.

In order to account for the changes of glycoforms related to ageing [35] and hormonal status [36] (e.g., menopause for women), two subsets with an age below or over 60 years old were considered (Appendix A). A complete overview of seven glycosylation traits for each IgG subclass (1, 2 and 4) is given in Figure 2. This highlighted the differences in the Fc glycosylation traits between the specified groups, which are often not of constant magnitude in the different IgG subclasses. As expected, Fc galactosylation of all IgG subclasses showed a diminished abundance in older subjects, especially in females, and even in the absence of a core fucose. A similar sex- and age-related behavior was observed for the decline of the sialylation of the antennas and the increase of bisected GlcNAc of all the three IgG subclasses with ageing (Figure 2).

### 2.2. IgG Glycan Signatures of IMN Patients

To explore the Fc glycosylation patterns of serum IgG associated with IMN, a comparison between IMN patients and age- and sex-matched healthy controls was performed. The statistical outcomes are summarized in Appendix A and in Figure 3.

In particular, IgG4 hypo-galactosylation was observed independent of the PLA2R level. Interestingly, a similar decline in IgG4 galactosylation was found even in the absence of the core fucose. A decrease of galactose levels was also observed for IgG1 and IgG2. However, these effects appeared less robust and were dependent on the matching approach applied.

Moreover, IMN was found to be significantly associated with an increase in hybrid type structures on IgG2 and the level of core fucosylation on IgG4. These observations remained significant with both matching approaches in all the comparisons for IgG2 hybrid-type glycans, whilst for IgG4 fucosylation in low PLA2R patients, this was only true with the clustered matching approach. Interestingly, the level of IgG4 hybrid type glycans was lower in IMN patients than in controls. This observation was made independently from the statistical approach used in the complete IMN dataset, as well as in the high PLA2R IMN subset.

The clustered gender-age matching also highlighted a decrease of bisected glycoforms linked to IgG2 and of the sialylation of IgG1 in the complete IMN patient set as compared to the controls.

### 2.3. IMN-Specificity of IgG Glycan Signatures

To explore whether the found glycosylation signatures were IMN-specific or rather a feature of general nephropathy, IgG Fc glycosylation was additionally studied in a sample set consisting of patients affected by a nephropathy not attributable to IMN, but with the same clinical presentation (nephrotic syndrome; *n* = 20).

Results are reported and summarized in Figure 4 and in Appendix A. Similar to the findings for the IMN patients, a significantly lower level of hybrid IgG 1 and 2 glycoforms was noted in PN, together with a decrease in IgG1 and 2 galactosylation. Furthermore, IgG1 sialylation was also lower in PN with respect to the healthy controls. Interestingly, none of the IgG4 glycosylation features were significantly altered between PN and healthy controls, whilst the levels of IgG4 galactosylation, hybrid type glycans, and fucosylation were significantly altered in the IMN patient group. This indicates the specificity of these IgG4 traits for IMN (Figure 5 and Figure 6).

## 3. Discussion

The exploration of the aberrant Fc glycosylation of IgG in IMN is motivated by the autoimmune character of this disease and the molecular processes at the base of its etiopathogenesis. Although several studies uncovered parts of the molecular mechanisms underlying the onset and the development of this nephrotic syndrome, none of them provided a systematic glycosylation analysis of blood immunoglobulins that could define the disease and simultaneously weighted and minimized the effect of interfering factors such as sex and age of the patients. Our approach not only investigated the mechanism of the disease, but it could also provide additional keys in the recognition and classification of the pathology. To increase the confidence of the outcome and to optimize the resolution of this IgG glycosylation map, both in terms of confounders and specificity, we performed extensive matching of interfering factors such as age and sex and investigated IgG glycosylation features in a pathological control group.

IgG functions are extensively and finely regulated through its interactions with receptors on immune cells and complement proteins. Alterations in these effector functions can be the results of a variation of immunoglobulin conformation and affinity for Fc gamma receptors (FcγRs), which is partly mediated by Fc glycans [37]. The diantennary glycan attached to Asn297 of IgG heavy chains can be decorated by 0 to 2 galactose and/or sialic acid residues on the branches. Furthermore, a large proportion of the IgG glycans carries a core-linked fucose, whilst some may carry a bisecting GlcNAc. Alternatively, a very low proportion of the circulating IgG molecules carry an immature hybrid-type glycan. In many pathological conditions the levels of the sialylation and/or galactosylation, as well as the presence of the core fucose and bisecting GlcNAc of IgG Asn297-linked glycans, is significantly altered [27]. Notably, several confounding factors have been demonstrated to influence the variability of IgG glycosylation independently of pathological status and introduce bias in analysis targeting IgG glyco-marker discovery [27]. The chronological and biological age of an individual, as well as sex-related hormonal effects, particularly related to the modulation of estrogen levels in women, were reported to extensively modify IgG glycosylation patterns [36,38]. Although other studies addressing this topic considered data from healthy subjects, no systematic and extensive analysis on these confounding factors specifically related to IMN has been reported thus far. Here, known age- and sex-related trends, such as the decline in IgG galactosylation and sialylation as well as the increase of bisecting N-glycans, were confirmed in both the healthy and diseased groups in the cohort. Galactosylation and sialylation have been shown to diminish with increasing age in a sex-specific manner, with a consistent drop in females around the age of 45 to 60 years, which is most likely attributable to menopause changes [28]. In a more recent study, the sex dependence of IgG N-glycosylation in blood was confirmed in a group of 669 healthy participants, pinpointing an increase of the core fucosylation level and the galactosylation level in females compared to those measured in males [39]. Although the exact mechanisms concealing age-related changes in IgG N-glycan profiles remains to be fully clarified, hypo- or agalactosylation of IgG was proposed to reflect biological ageing and to enhance the pro-inflammatory potential of IgG [40]. This glycomic alteration has been reported in association with a low-grade, chronic, asymptomatic inflammatory state that can anticipate and sustain disease onset and often follows remission, e.g., in rheumatoid arthritis [32].

The specificity of the glycosylation patterns we found to be associated with IMN was validated based on their further investigation in a cohort of patients suffering from kidney disease different from IMN. The pathological control group was chosen on the basis of a clinical presentation (nephrotic syndrome) similar to that of IMN. However, the natural age and sex distribution between PN and IMN patients is large, which was also reflected in our sample set. Therefore, a direct comparison between PN and IMN was not possible. Instead, both patient groups (IMN and PN) were compared against an age- and sex- matched subset of the healthy controls, based on two different matching procedures. In both sample sets, we found the levels of IgG1 and 2 hybrid type glycans to be higher with respect to healthy controls, whilst the galactosylation levels of these subclasses was lower in the diseased individuals. This suggested that these features are connected to the general clinical phenotype of the diseases, such as renal damage or failure, or inflammation, rather than to the presence of a specific disease. Indeed, aberrant IgG N-glycan profiles have already been observed in other non-IMN nephropathies. A decrease of IgG core fucosylation was noted to be lowered in systemic lupus erythematosus (SLE) [41]. Moreover, a very recent study involving a publicly available database of genome-wide association studies of IgG N-glycosylation and disease risks determined a positive effect of SLE on the abundance of N-glycans with bisecting GlcNAc in the total IgG N-glycome [42]. Barrios and Menni et al. demonstrated that 14 different IgG glycan traits belonging to galactosylation, sialylation, and bisecting features were associated with kidney function [21]. Kao et al. showed that IgG1 sialylation was associated with eGFR (Estimated Glomerular Filtration Rate) values, confirming that the decrease of sialylated IgG is an indicator of the inflammatory status of individuals with lower kidney function [43].

IgG glycosylation traits that were specific to IMN as compared to related pathologies were mainly IgG4-specific (Figure 5 and Figure 6), showing a lower abundance of hybrid type glycans and an increase of core fucosylation. Additionally, the hypo-galactosylation of IgG4 was associated exclusively with IMN, even for the species without the core fucose. Finally, the increase of the sialylation on IgG4 and a lower level of bisecting GlcNAc on IgG2 also seemed specific to IMN patients.

Interestingly, none of the disease signatures shared between IMN and PN involved IgG4 (Figure 6). This data appears to be in line with the known serum ratio of IgG4 in patients with membranous nephropathy [31]. A common finding in IMN is that autoantibodies belong largely to the IgG4 subclass, in contrast to other nephrotic syndromes like systemic lupus erythematosus (SLE), in which IgG1, IgG2, and IgG3 are prevalent [44]. Although IgG4 is known to not activate the classic complement pathway, abundant renal deposits of complement, including C4 (but not C1q) have been shown in IMN patients, suggesting a specific role of IgG4 in the pathogenesis of this disease [13,31].

To our knowledge, among these aberrations, IgG4 galactose deficiency was the only one to be previously associated with IMN. A galactosylation in total serum and on antigen-specific IgG is widely known to have pro-inflammatory implications and is suggested to be involved in various chronic inflammatory and autoimmune pathologies via the activation of the alternative complement pathway and antibody-mediated phagocytosis [20]. Terminal galactoses are possibly involved in the activation of complement-dependent cytotoxicity (CDC), through C1q complement component binding [17]. Very recently, Haddad et al. showed in an in vitro model that anti-PLA2R IgG4 autoantibodies can activate the lectin complement pathway, prompting a sublethal injury of PLA2R-expressing podocytes and that this effect is facilitated by an aberrant galactose deficiency of anti-PLA2R IgG4 [15]. Interestingly, our data support these observations and, in addition, they highlight that this alteration occurs additionally in the absence of a core fucose. The lack of this trait is known to increase ADCC (antibody-dependent cellular cytotoxicity), affecting the binding affinity for FcRIIIa and FcRIIIb by up to 100 times [18,19]. This effect was largely studied and has brought about the successful development and approval for the next-generation of glycoengineered therapeutic monoclonal antibodies (mAbs) with low fucose levels, such as Gazyva (obinutuzumab, Roche/Genentech) [45]. Moreover, we reported a significant increase in IgG4 fucose levels. Recently, a study performed on 101 consecutive treatment-naïve SLE patients with positive anti-dsDNA antibodies suggested a similar increase in the fucosylation of anti-dsDNA IgG1 that was correlated with SLE disease activity (SLEDAI scores) [46]. Finally, our data showed a lower abundance of IgG4 Fc hybrid-type glycans in patients affected by IMN. The physiological role of these glycans is not completely elucidated and there is no fixed tendency regarding their blood expression in pathological processes. Higher serum levels of these IgG glycans were found in colorectal cancer, hepatocellular carcinoma, and in non-alcoholic steatohepatitis, but not in pancreatic ductal carcinoma [47,48,49]. In G4-related disease (IgG4-RD), a reduction in hybrid structures was reported, which was reversed upon steroid treatment, and correlated with disease activity. In particular, Culver et al. reported also that IgG1/2/3 hybrid structures negatively correlated with complement C3 and C4 levels in this systemic fibro-inflammatory condition [50].

The only IMN-specific glycosylation trait observed for an IgG subclass other than IgG4 was the level of bisecting GlcNAc on IgG2. The presence of this GlcNAc was shown to increase FcɣRIIIa affinity independent of afucosylation and galactosylation [51]. Moreover, increased bisection has been reported in patients with chronic kidney disease [21], as well as in other nephropathies [52]. Recently, Magorivska et al. reported a glycoproteomic study in which seropositive rheumatoid arthritis was accompanied by the appearance of IgG glycans bearing a bisecting GlcNAc residue [53]. Of note, the clustered matching strategy also showed a higher degree of IgG4 sialylation per galactose in IMN patients, which was not observed in the PN versus control comparison. The addition of a terminal sialic acid to IgG N297 glycans has been shown to decrease the affinity for type I Fc receptors and increase the affinity for type II Fc receptors [54]. Moreover, the percentage of diantennary sialylated IgG has also been shown to decline in individuals with chronic kidney disease [21].

Some of the IMN-specific IgG4 glycosylation features seemed to be slightly more pronounced in the individuals with high PLA2R Ab titers (Figure 5 and Figure 6). However, the direct comparison of these traits between high and low PLA2R Ab patients did not result in any significant findings (data not shown), which is probably related to a mismatch in sample group characteristics and should be investigated further. Serum levels of PLA2R antibody have been used in IMN patients for diagnostic and prognostic purposes [55,56]. Furthermore, PLA2R Ab positivity has been correlated with more severe clinical symptoms and a poorer prognosis, which is most likely attributed to increased IgG4 deposition [57]. In our data, IgG4 hypo-galactosylation was very constant in its effect over the different PLA2R-titer groups, being in line with the observations of Haddad et al. that the aberrations in IgG glycosylation pattern, particularly the galactose-deficiency, were not only peculiar to PLA2R-specific autoantibodies [15].

The implications of our findings appear to have a double value. On one hand, the recognition of a typical Fc glyco-identity of IMN could be attractive for biological purposes, enabling the pathological molecular mechanisms to be better clarified. On the other hand, it could provide useful clinical information, uncovering the variability of the disease. In this perspective, once validated and accompanied by an investigation on the range of IgG Fc-glycan heterogeneity in different ethnic populations, our data can be used both for enriching the diagnostic spectrum as well as to track the variability of patients, helping to drive their classification.

## 4. Materials and Methods

### 4.1. Chemicals

Formic acid, ammonium bicarbonate, and TPCK (tosylsulfonyl phenylalanyl chloromethyl ketone)-treated trypsin from bovine pancreas were obtained from Sigma-Aldrich (Steinheim, Germany). SupraGradient acetonitrile (ACN) was obtained from Biosolve (Valkenswaard, Netherlands), and ultrapure deionized water (MQ) was generated by Purelab Ultra, maintained at 18.2 M (Veolia Water Technologies Netherlands BV, Ede, Netherlands). Trifluoroacetic acid was purchased from Merck (Darmstadt, Germany).

### 4.2. Human Sera Collection

Patients with a histological diagnosis of IMN and that were positive for PLA2R Ab in the serum, as well as patients affected by a nephropathy not attributable to IMN but with the same clinical presentation (nephrotic syndrome; PN), were included. All the patients were Caucasian. The patients were enrolled at time of diagnosis. Their serum was collected at the institutions where they were diagnosed and followed. Samples were collected, processed, and tested immediately or stored at −80 °C until testing. The diagnosis of IMN and clinical evaluation of the disease activity were performed by the physician in charge of the study in each of the participating nephrology units. The study was approved by the local ethics committee (N. 150/ST/2014, 1 October 2014 and informed consent was obtained from all patients for the treatment of data already collected for routine clinical use. Patient data were anonymously analyzed in accordance with the latest version of the Helsinki Declaration of human research ethics. Serum samples of healthy controls, selected to be age- and sex-matched to patients, were collected in accordance with the Declaration of Helsinki and Good Clinical Practice guidelines).

Blood samples were collected using a 5 mL serum-collection tube. After keeping the samples for 30 min at room temperature in an upright position, they were centrifuged for 15 min at 15,000 rpm. Aliquots of the serum were then stored at −80 °C until analysis.

Only anti-PLA2R-positive IMN patients were included in the cohort. Anti-PLA2R antibody detection was performed using the commercial Human Embryonic Kidney (HEK)-293 transfected cell-based IIFT (PLA2R IIFT, Euroimmun), as previously described [58,59]. The system consisted of glass slides containing two different biochips for each well, one coated with the PLA2R-1 c-DNA-transfected cells (substrate, overexpressing the full-length human PLA2R1) and the other with “mock” transfected HEK-293 cells as negative controls (mock transfection is transfection without DNA and is performed to control the potential impact of the transfection reagent on the cells). Briefly, samples from patients and controls were diluted in phosphate buffered saline (PBS) and incubated over the slide wells for 30 min at room temperature. After discarding and washing the unbound material, slides were overlaid with the FITC-conjugate (Fluorescein Isothiocyanate Conjugate) goat anti-human IgG for a further 30 min, washed again, and covered for the evaluation with a UV-equipped microscope (Leica DM 1000, light source mercury vapor lamp, excitation filter 490 nm and emission filter 520 nm, 100 W, 40× magnification) (Leica Microsystems, Heidelberg, Germany). The presence of fluorescent staining on the cytoplasm of the PLA2R transfected cells, partly including the cell membrane, was considered positive; a negative pattern of the control biochip was used as confirmation. For IMN patients, a high level of antibodies against PLA2R was referred to subjects showing a related titer cut-off >999.

### 4.3. IgG Capturing and Trypsin Digestion

Sera from 77 different clinical samples and 88 healthy control samples were randomized in two 96-well plates, together with 12 VisuCon pooled plasma standards (Affinity Biologicals, Inc., Ancaster, ON, Canada [6 per plate]), 6 PBS blanks (3 per plate), and 9 sample replicates. A controlled randomization with a uniform distribution of age, sex, and case/control ratio per plate was performed, as seen in Appendix A. IgGs were captured using ProtA affinity beads (GE Healthcare rProtein A Sepharose Fast Flow) in order to highlight glycopeptides deriving from IgG4 subclasses, with a minimal signal overlapping given by IgG3, based on a previously described protocol with slight modifications [60,61]. Briefly, 2 µL of plasma diluted in 40 µL of PBS was incubated with 2 µL of beads suspended in PBS for each sample for 1 h with agitation. Incubation was followed by three washing steps with 200 µL of PBS and three with 200 µL of MQ water. The elution of antibodies was performed with 100 µL of 100 mM formic acid. Samples were then dried for 3 h at 60 °C in a vacuum concentrator and dissolved in 20 µL of 25 mM ammonium bicarbonate. After 10 min of shaking, 20 µL of trypsin solution was added (500 ng/sample). Samples were incubated at 37 °C for 18 h.

### 4.4. LC-MS Analysis of IgG Glycopeptides

The glycopeptide mixtures were separated and analyzed using an Ultimate 3000 high-performance liquid chromatography (HPLC) system (UltimateTM 3000 RSLCnano, ThermoFisher Scientific, Sunnyvale, CA, USA) coupled online to an Impact HD Ultra High-Resolution flight mass spectrometer (UHR-TOF) (Bruker Daltonics, Bremen, Germany), as already reported [60]. Briefly, 0.2 µL of each sample-well of the plates placed in the autosampler at 4 °C was injected and subjected to concentration and desalting into a trap column (Dionex Acclaim PepMap100 C18, 5 mm × 300 μm; Thermo Fisher Scientific, Breda, Netherlands). Glycopeptides were separated on a 10 cm analytical column BEH C18 (nanoEase *m*/*z* peptide, 130 A, 1.7 µm, 75 µm × 100 mm, Waters) with a flow rate of 0.6 µL/min at 45 °C using a 5 min multi-step gradient from 3 to 50% of solvent B (95% Acetonitrile). The mass spectrometry (MS) device was equipped with CaptiveSpray and nanoBooster technologies (Bruker Daltonics), using ACN-doped nebulizing gas. Spectra were recorded in profile mode within the *m*/*z* range 550 to 1800 using a frequency of 1 Hz with the following tune parameters: collision energy of 7.0 eV, transfer time of 110 μs, and a e pre-pulse storage of 21 μs. The total runtime was approximately 11.5 min for each sample.

### 4.5. Data Pre-Processing

Raw data derived from LC-MS analysis were processed using the platform LacyTools v1.1.0 (Build 20180508a) as already described [62]. Briefly, glycopeptides belonging to the same IgG isoform (IgG1, IgG4, or IgG2) were clustered together. Alignment based on the exact mass was performed using 4 s and 0.1 Th as time and *m*/*z*, respectively, using at least 7 features with a signal-to-noise ratio (S/N) greater than 100. Aligned sum spectra were then calibrated on five peaks (S/N ≥ 9). For the extraction, an analyte list was built including all glycoforms for each sub-classes. The resolution of extracted analytes was accepted if there were above 100 data points per 1 *m*/*z* for features with a charge from +2 to +4 in a 12 sec time window. Analytes were included if the following parameters were satisfied: their average S/N ratio was greater than 9, their isotopic pattern did not, on average, deviate more than 20% from the theoretical pattern, and their average mass error was within ±10 ppm. This resulted in the extraction of 56 features, divided in 21 for IgG1, 18 for IgG2, and 17 for IgG4 glycoforms (Appendix A). Low-intensity spectra were excluded when the total glycopeptide signal intensity was below the average minus three times the standard deviation. This resulted in the exclusion of one IgG, one IgG2, and three IgG4 spectra. The absolute intensities of the extracted glycoforms were total area normalized per IgG subclass and glycosylation traits were calculated based on specific glycosylation features, as detailed in Appendix A [63].

### 4.6. Statistical Analysis

For this study, we considered two matching designs with a fixed and a variable number of cases and controls: 1:1 (1 case: 1 healthy control) and m:n (m cases: n healthy controls; clustered matching). Each IMN case was matched to 1 or all available healthy controls, according to the following criteria: sex and age (±3 years).

Descriptive statistics were reported as absolute frequencies and percentages or median and quartiles (Q1–Q3), for categorical and continuous variables, respectively. The Clustered Wilcoxon test, that accounts for the matched design, was used to compare glycoproteins between groups. The Hochberg method was considered to adjust for multiple testing and significance tests were reported in terms of adjusted *p*-values. For each test, the level of significance was set equal to 5%. Statistical analyses were performed using the open-source R software v.3.6.0 (R Foundation for Statistical Computing, Vienna, Austria). 

## 5. Conclusions

Our work highlighted a typical Fc glycosylation pattern of IgG1, 2, and 4 associated with the IMN, weighting and minimizing the interfering effects of age and sex and evaluating its specificity in terms of the clinical presentation of the disease. Interestingly, the specific signatures connected to IMN were IgG4-related. In particular, the hypo-galactosylation of IgG4, already reported for IMN, was confirmed, and our data supported that this alteration occurs even in the absence of the core fucose, suggesting an enhancement of the pro-inflammatory potential of IgG4. Furthermore, a higher level of fucosylation on IgG4, along with a lower IgG4 hybrid-type glycan level, was reported to be specific for IMN.

These results are potentially useful to uncover the molecular mechanisms of the disease and represent a starting point for improving the classification of patients with IMN, and thus, therefore help guide the most appropriate therapeutic treatment.

## Figures and Tables

**Figure 1 ijms-23-04664-f001:**
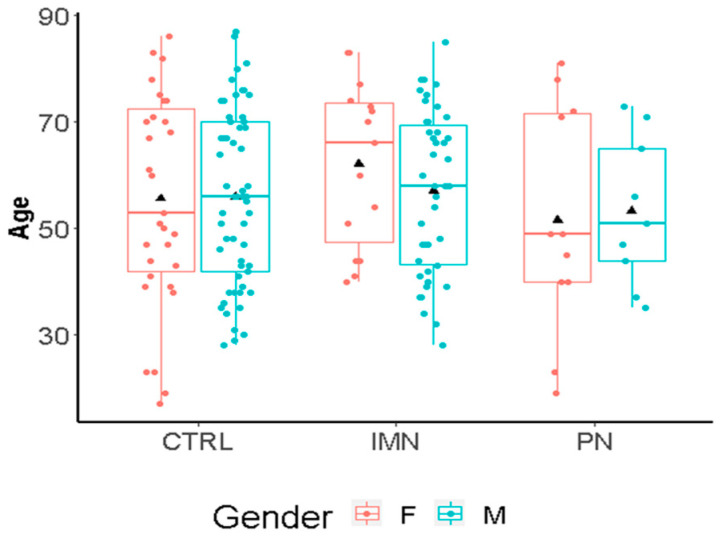
Age distribution by gender and groups (IMN = Idiopathic Membranous Nephropathy; CTRL = healthy controls; PN = pathological controls with non-IMN nephropathy). Age refers to years. The black triangle represents the average.

**Figure 2 ijms-23-04664-f002:**
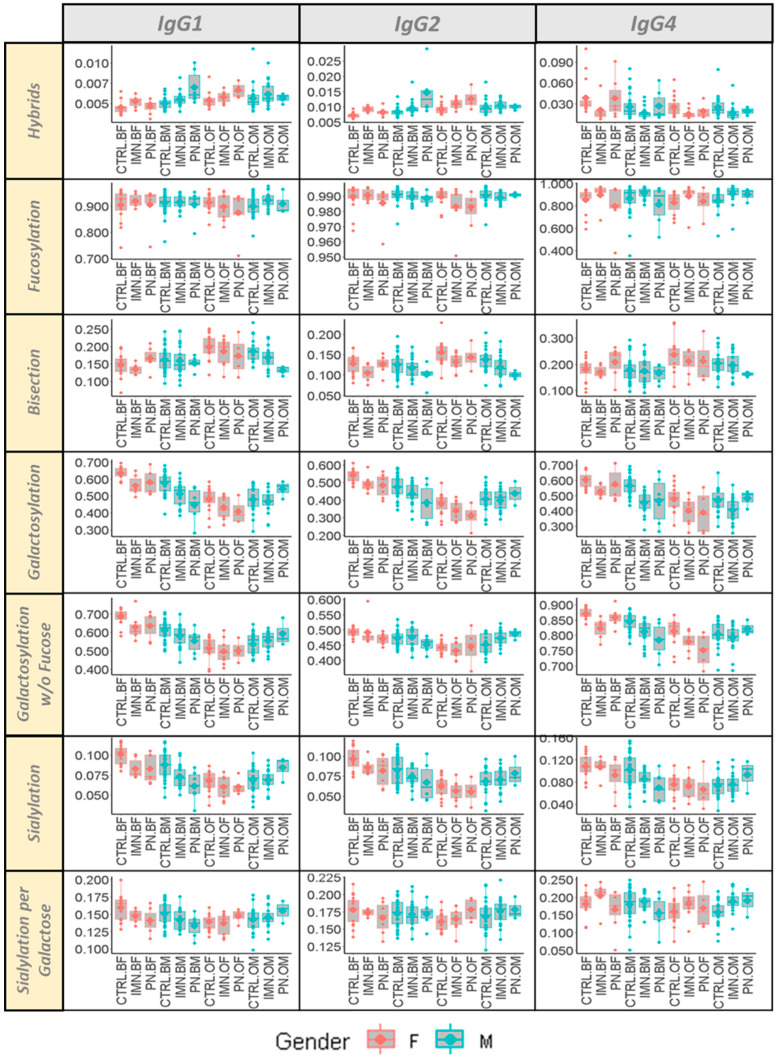
Box plots of seven glycosylation traits of IgG1, IgG2, and IgG4 stratified according to sex (F = female, M = males) and age (O = over 60 years old—age ≥ 60; B = below 60 years old—age < 60) for all the groups of the cohort (IMN = Idiopathic Membranous Nephropathy; CTRL = healthy controls; PN = pathological controls with non-IMN nephropathy).

**Figure 3 ijms-23-04664-f003:**
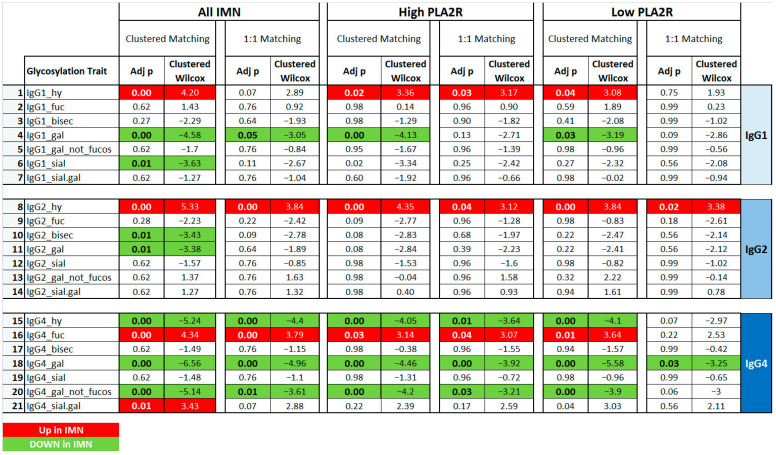
Comparisons between IMN patients and matched controls considering the seven glycosylation traits* for each IgG subclass (1, 2 and 4). Matching was performed using a **clustered** or **one-to-one** approach. The IMN patients were used all together (**ALL IMN**) or based on PLA2R titer (**HighPLA2R** if > 999; **Low PLA2R** if < 999). **Green and bold** indicate significant (*p* < 0.05) lower levels In IMN patients, while **red and bold** refers to significant (*p* < 0.05) higher levels in IMN patients. ** hy = hybrid glycoforms; fuc = with fucosylation; bisec = with bisection; gal = with galactosylation; sial = with sialylation; gal_not fucos = galactosylation of non-fucosylated glycoforms; sial.gal = sialylation per galactose*.

**Figure 4 ijms-23-04664-f004:**
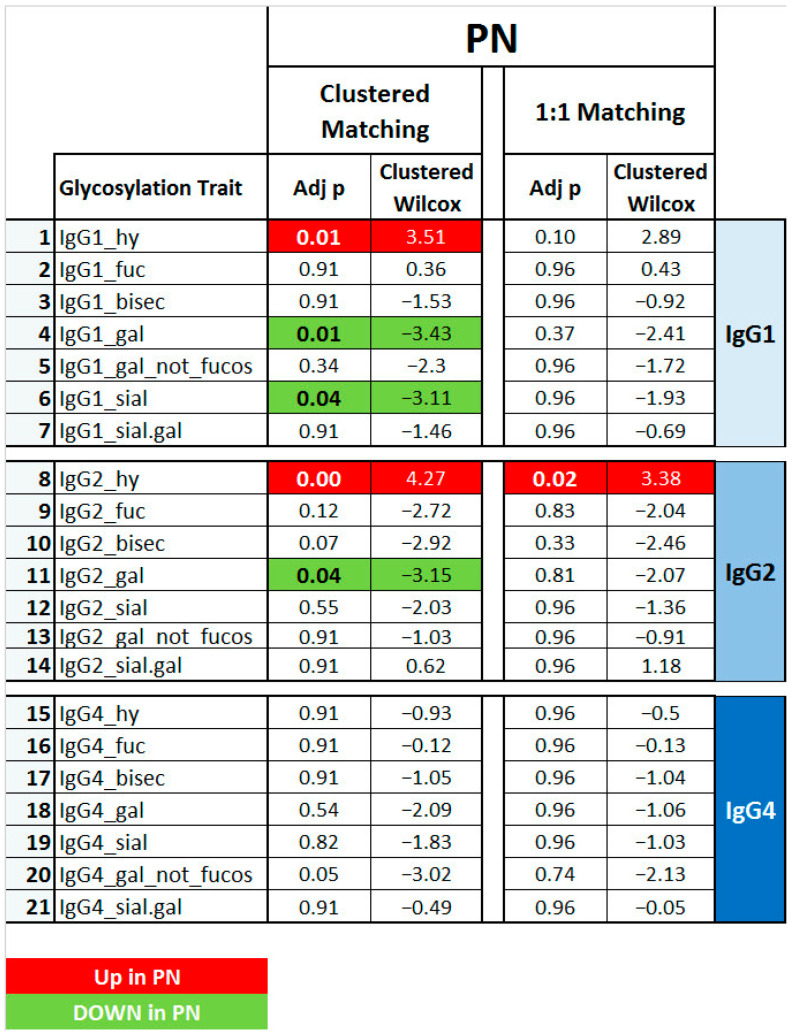
Comparisons between PN patients and matched controls considering the seven glyco-patterns* for each IgG subclass (1, 2 and 4). Matching was performed using a **clustered** or **one-to-one** approach. **Green and bold** indicate significant (*p* < 0.05) lower levels in PN patients, while **red and bold** refers to significant (*p* < 0.05) higher levels PN patients (clustered Wilcox). ** hy = hybrid glycoforms; fuc = with fucosylation; bisec = with bisection; gal = with galactosylation; sial = with sialylation; gal_not fucos = galactosylation of non-fucosylated glycoforms; sial.gal = sialylation per galactose*.

**Figure 5 ijms-23-04664-f005:**
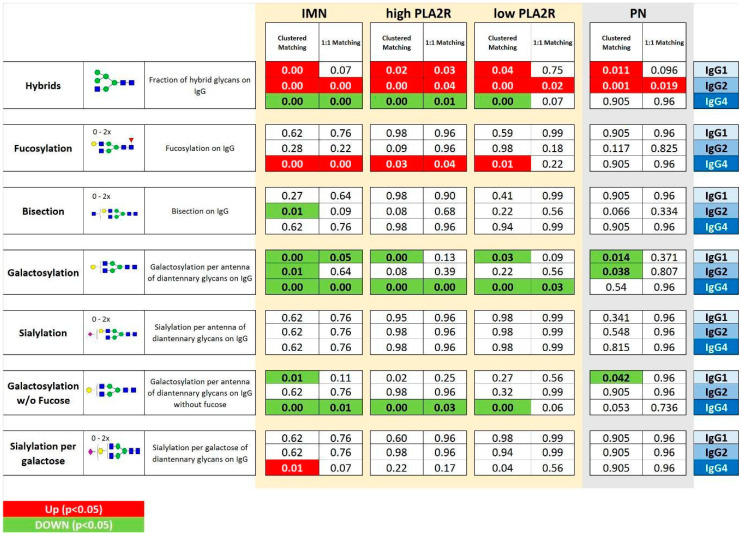
Overview of IgG Fc glycosylation signatures of idiopathic membranous nephropathy (**IMN**) and other nephropathies that are different from IMN (**PN**), compared in both cases to healthy controls. The *p*-values are given for seven N-glycosylation traits of serum IgG1, 2, and 4. Significant differences (*p* < 0.05) are highlighted in **bold** and colored based on the direction of the difference (**green** for lower values and **red** for higher values in disease vs. control). The IMN patients were used all together (**IMN**) or based on PLA2R titer (**HighPLA2R** if > 999; **Low PLA2R** if < 999). For the comparisons, sex- and age-matching with healthy controls were performed based on the clustered and one-to-one approach.

**Figure 6 ijms-23-04664-f006:**
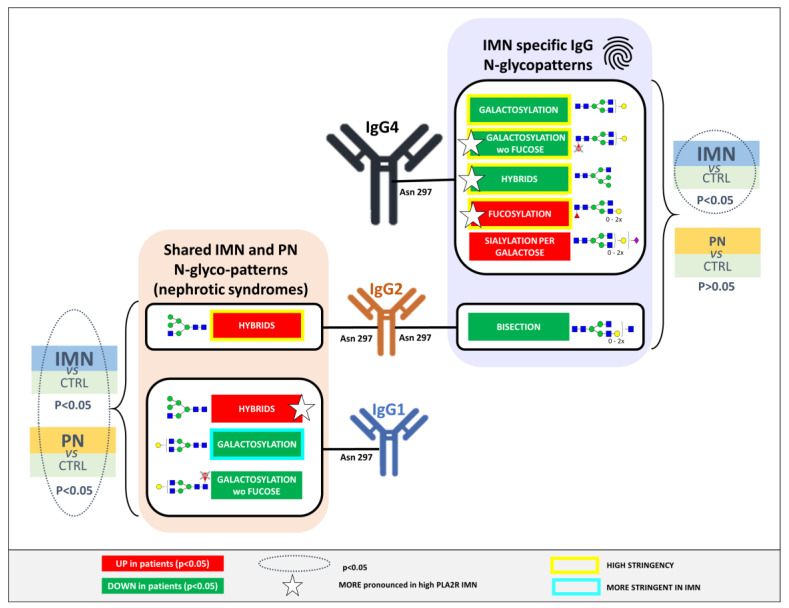
Illustration of the significant *N*-glycan signatures of serum IgG1, 2, and 4 associated specifically to IMN or both IMN and pathological controls. Features indicated with a star were more pronounced in high PLA2R Ab titer IMN patients. The stringency (yellow/blue box) refers to the significance considering both matching approaches.

## Data Availability

The data presented in this study are available on request from the corresponding author.

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
