# Peer review of "Definition of IgG Subclass-Specific Glycopatterns in Idiopathic Membranous Nephropathy: Aberrant IgG Glycoforms in Blood"

_ijms, 2022, doi:10.3390/ijms23094664_

Round 1

Reviewer 1 Report

The authors seek to define IgG subclass specific glycopatterns in IMN in order to help with prognostication/potential future treatment options. The study uses mass spectrometry as the main tool. it provides some information consistent with what is already in the literature, but also some new information to help with understanding the disease better.

1) Introduction: This glomerular disease can present a primary (idiopathic) or secondary aetiology.
In developed countries, approximately 75% of all MN are idiopathic (or primary); the remaining 20–25% are secondary to different conditions, such as infections (hepatitis B virus, hepatitis C virus, human immunodeficiency virus, malaria, etc.), malignancies (lung,
breast, stomach and ovarian cancer, lymphoproliferative disorder, etc.), systemic autoimmune disease (lupus, rheumatoid arthritis, etc.) and drugs [3]. - this statement is now inaccurate - the classification is now an antigen based classification and this should be clarified..

2) Those with other nephropathy is much smaller at n=20 compared to the MN group n=57 - why not increase to make it comparable? Also the control group is not similar in terms of gender.

3) Patient's were controlled for age and gender, but not for ethnicity/race - this is an important thing to control for as MN is much more prevalent in those of causcasian race but affects other groups as well.

4) The authors report patients were classified as high PLA2R titer vs Low, but do not report what cut-off they used -this is important information to include for the reader given so much is spent on discussing differences between high or low patients.

Author Response

The authors seek to define IgG subclass specific glycopatterns in IMN in order to help with prognostication/potential future treatment options. The study uses mass spectrometry as the main tool. it provides some information consistent with what is already in the literature, but also some new information to help with understanding the disease better.

1) Introduction: This glomerular disease can present a primary (idiopathic) or secondary aetiology.
In developed countries, approximately 75% of all MN are idiopathic (or primary); the remaining 20–25% are secondary to different conditions, such as infections (hepatitis B virus, hepatitis C virus, human immunodeficiency virus, malaria, etc.), malignancies (lung,
breast, stomach and ovarian cancer, lymphoproliferative disorder, etc.), systemic autoimmune disease (lupus, rheumatoid arthritis, etc.) and drugs [3]. - this statement is now inaccurate - the classification is now an antigen based classification and this should be clarified.

We thank the reviewer for the suggestion. Accordingly, we have elucidated this aspect concerning the proposed antigen-based classification in the introduction.

2) Those with other nephropathy is much smaller at n=20 compared to the MN group n=57 - why not increase to make it comparable? Also the control group is not similar in terms of gender.

We agree with the reviewer that the control group was quite small. However the reasons, underlying this issue, are ascribable to the rarity of the glomerular disease and also to the selection of patients based on the nephrotic range proteinuria. All the differences in terms of gender and age were minimised using the strictly matched healthy controls as normalising factor. This point has been better specified in results section 3.3. 

3) Patient's were controlled for age and gender, but not for ethnicity/race - this is an important thing to control for as MN is much more prevalent in those of causcasian race but affects other groups as well.

Thanks for highlighting an important factor for MN. Patients were not controlled for ethnicity/race because they were all caucasian. This point has been clarified in the text.

4) The authors report patients were classified as high PLA2R titer vs Low, but do not report what cut-off they used -this is important information to include for the reader given so much is spent on discussing differences between high or low patients.

Thanks for noticing. We have specified this aspect also in the text (Results 3.1) and not only in the captions.

The manuscript was edited by an English speaking colleague, and submitted to a general restyle to reduce possible redundancies. All the supplemental material was separated in a different file to visualise better the key findings

Reviewer 2 Report

In the submitted article, the authors attempted to define IgG subclasses-specific glycopatterns in Idiopathic membranous nephropathy. The manuscript is well written and the novelty is clearly explained. All findings are supported by clear tables and figures. The discussion is at a high scientific level. One of the major benefits of this work is the well-described experimental part. In my opinion, the manuscript is in accordance with the scope of the International Journal of Molecular Science and is interesting for readers in the field of analysis of biological important analytes. For this reason, I recommend to accept this manuscript.

Author Response

The manuscript was edited by an English speaking colleague, and submitted to a general restyle to reduce possible redundancies. All the supplemental material was separated in a different file to visualise better the key findings

Reviewer 3 Report

The authors describe an important study that illustrates the possible relevance of detecting variant IgG glycosylation patterns for the diagnosis/treatment of idiopathic membranous nephropathy (IMN).  However, there are some issues that must be addressed prior to publication of this paper.  They are as follows: 

(1) Within the Discussion section (on page 21) there appears to be a misstatement regarding the glycopatterns shared between PN and IMN.   The authors state that "only two glyco-patterns signficantly associated to IMN...were shared in a consistent manner, as a decrease of IgG2 hybrid forms and the hypo-galactosylation in IgG1 (Figure 3)."  However, according to Figure 3, there appears to be an INCREASE in IgG2 hybrid forms, not a decrease, and this feature seems to be shared with IgG1 as well (looking at the clustered matching data).  Therefore, this statement must be corrected and modified accordingly.

(2) On page 23 of the Discussion section, a reference is made to the type of linkage of the sialic acid (e.g. α2,3- vs. α2,6-).  However, it was not made clear in any of the tables or figures which linkage(s) was(were) found in the samples.  This should be clarified early on.  And if linkage wasn't determined, the discussion of sialic acid linkage should be altered to indicate that this is something to be looked at in a future study.

(3) The formation mechanism and the role of anti-PLA2R antibodies in IMN should be further elaborated on in the Introduction section so that the reader can better understand the reasoning behind studying these specific antibodies.  The reference to Beck et al. is a good start, but not sufficient.  

(4) It is unclear why there are only supplemental data for the results described in section 3.3.

(5) In Figure 1, the X and Y axes are extremely difficult to read.  This needs to be corrected.  In addition, though there are different colors on the box plot, these colors are not defined in the figure legend.  This, too, must be fixed.

(6) It would be most helpful to have a final figure (perhaps in the Discussion) illustrating the differences in the IgG4 glycosylation pattern between IMN patients and healthy individuals and/or between IMN and PN patients.  A large illustration depicting the ≥8 sugars on a "healthy" IgG4 vs. "IMN" IgG4 would clarify the findings for the reader.

(7)  It is strongly suggested that the entire document be proofread by someone who is quite fluent in English, specifically, someone who is an expert at writing in English.

(8) The following are minor issues that should be addressed prior to publication:

             a) In the Abstract, the phrase "Several IgG Fc glyco-signatures of        IMN met all the selection criteria" is stated twice.

             b) The subheading "Introduction" should be preceded by the number 1.

              c) Make sure all acronyms/abbreviations are clearly defined the first time they are used.

              d) Table titles should be ABOVE the table, not below. Definitions of acronyms/abbreviations can be placed below the table.

              e) Figure S2 is really a table, not a figure.  In addition, there is a typo in one of the headings-- "OB" should be replaced with "OM."

Author Response

The authors describe an important study that illustrates the possible relevance of detecting variant IgG glycosylation patterns for the diagnosis/treatment of idiopathic membranous nephropathy (IMN).  However, there are some issues that must be addressed prior to publication of this paper.  They are as follows: 

(1) Within the Discussion section (on page 21) there appears to be a misstatement regarding the glycopatterns shared between PN and IMN.   The authors state that "only two glyco-patterns signficantly associated to IMN...were shared in a consistent manner, as a decrease of IgG2 hybrid forms and the hypo-galactosylation in IgG1 (Figure 3)."  However, according to Figure 3, there appears to be an INCREASE in IgG2 hybrid forms, not a decrease, and this feature seems to be shared with IgG1 as well (looking at the clustered matching data).  Therefore, this statement must be corrected and modified accordingly.

We thank the reviewer for noticing this error. We corrected the statement according to the observation. 

(2) On page 23 of the Discussion section, a reference is made to the type of linkage of the sialic acid (e.g. α2,3- vs. α2,6-).  However, it was not made clear in any of the tables or figures which linkage(s) was(were) found in the samples.  This should be clarified early on.  And if linkage wasn't determined, the discussion of sialic acid linkage should be altered to indicate that this is something to be looked at in a future study.

We agree with the reviewer that this point was not adequately developed and can be a source of  misinterpretations. Since the specific linkage of the sialic acid has not been determined, and not useful for IgG, we choose to delete the citation.

 (3) The formation mechanism and the role of anti-PLA2R antibodies in IMN should be further elaborated on in the Introduction section so that the reader can better understand the reasoning behind studying these specific antibodies.  The reference to Beck et al. is a good start, but not sufficient.

Thanks for raising this topic. Indeed, even if there is a consensus of using anti-PLA2R antibodies as a diagnostic and prognostic biomarker, many issues addressing initiation of antibody formation  and the timeline of their production along with the role of IgG subclasses and antigenic epitopes, remain unresolved. A digression about this theme and how autoimmunity develops in MN has been added in the introduction.

(4) It is unclear why there are only supplemental data for the results described in section 3.3.

We agree with the reviewer and accordingly we move the related figure about the p-values of PNvs CTRLs comparison into results from the supplemental material (Figure S3 changed in Figure 4).

(5) In Figure 1, the X and Y axes are extremely difficult to read.  This needs to be corrected.  In addition, though there are different colors on the box plot, these colors are not defined in the figure legend.  This, too, must be fixed.

According to the referee's suggestion, we have modified Figure 1 (now Figure 2).

  • (6) It would be most helpful to have a final figure (perhaps in the Discussion) illustratingthe differences in the IgG4 glycosylation pattern between IMN patients and healthy individuals and/or between IMN and PN patients.  A large illustration depicting the ≥8 sugars on a "healthy" IgG4 vs. "IMN" IgG4 would clarify the findings for the reader.

Thanks for the very useful suggestion. We have inserted a Figure in Discussion section (Figure 6) in order to make the outcome of the Igg Glycosylation alterations clearer and more easily accessible.

(7)  It is strongly suggested that the entire document be proofread by someone who is quite fluent in English, specifically, someone who is an expert at writing in English.

According to the reviewer, the manuscript was edited by an English-speaking colleague, and submitted to a general restyle to reduce possible redundancies. All the supplemental material was separated in a different file to visualise better the key findings

(8) The following are minor issues that should be addressed prior to publication:

  1.           a) In the Abstract, the phrase "Several IgG Fc glyco-signatures of        IMN met all the selection criteria" is stated twice. 
  2.            b)The subheading "Introduction" should be preceded by the number 1. 
  3.           c)Make sure all acronyms/abbreviations are clearly defined the first time they are used. 
  4.     d)Table titles should be ABOVE the table, not below. Definitions of acronyms/abbreviations can be placed below the table.
  5.             e)Figure S2 is really a table, not a figure.  In addition, there is a typo in one of the headings-- "OB" should be replaced with "OM." 

All the points have been addressed following the reviewer’s suggestions.